# Predicting Self-Medication with Cannabis in Young Adults with Hazardous Cannabis Use

**DOI:** 10.3390/ijerph19031850

**Published:** 2022-02-07

**Authors:** Dorothy Wallis, J. Douglas Coatsworth, Jeremy Mennis, Nathaniel R. Riggs, Nikola Zaharakis, Michael A. Russell, Aaron R. Brown, Stephanie Rayburn, Aubrie Radford, Christopher Hale, Michael J. Mason

**Affiliations:** 1Center for Behavioral Health Research, College of Social Work, The University of Tennessee Knoxville, 1618 Cumberland Avenue, Knoxville, TN 37996, USA; nzaharak@utk.edu (N.Z.); chale18@utk.edu (C.H.); 2Department of Geography and Urban Studies, Temple University, Philadelphia, PA 19122, USA; jmennis@temple.edu; 3CSU Prevention Research Center, Colorado State University, Fort Collins, CO 80523, USA; riggsnr@colostate.edu (N.R.R.); aubrie.radford@colostate.edu (A.R.); 4Biobehavioral Health, Penn State College of Health and Human Development, University Park, State College, PA 16802, USA; mar60@psu.edu; 5Department of Social Work, Western Carolina University, Cullowhee, NC 28723, USA; brownaaron@wcu.edu; 6Department of Human Development and Family Studies, Colorado State University, Fort Collins, CO 80523, USA; stephanie.rayburn@colostate.edu

**Keywords:** anxiety, cannabis use, cannabis use disorder, depression, self-medication, withdrawal symptoms, young adults

## Abstract

Using cannabis to reduce psychological and physical distress, referred to as self-medication, is a significant risk factor for cannabis use disorder. To better understand this high-risk behavior, a sample of 290 young adults (ages 18–25; 45.6% female) were recruited from two U.S. universities in January and February of 2020 to complete a survey about their cannabis use and self-medication. Results: seventy-six percent endorsed using cannabis to reduce problems such as anxiety, sleep, depression, pain, loneliness, social discomfort, and concentration. When predicting reasons for self-medication with cannabis, logistic regression models showed that lower CUDIT-R scores, experiencing withdrawal, living in a state where cannabis was illegal, and being female were all associated with higher rates of self-medication. Withdrawal symptoms were tested to predict self-medication with cannabis, and only insomnia and loss of appetite were significant predictors. To further explore why young adults self-medicate, each of the original predictors were regressed on seven specified reasons for self-medication. Young adults experiencing withdrawal were more likely to self-medicate for pain. Participants living where cannabis is legal were less likely to self-medicate for anxiety and depression. Living where cannabis is illegal also significantly predicted self-medicating for social discomfort—though the overall model predicting social discomfort was statistically non-significant. Finally, female participants were more likely to self-medicate for anxiety. These results suggest widespread self-medication among young adults with likely CUD and underscore the complexity of their cannabis use. The findings have implications for understanding why young adults use cannabis in relation to psychological and physical distress and for accurately treating young adults with cannabis use disorder.

## 1. Introduction

Young adulthood (ages 18 to 25) is a critical developmental period for onset and increase in cannabis use, with lifetime use rates more than tripling between adolescence (15.8%) and young adulthood (51.7%) in the U.S. [1]. Both frequent and sustained cannabis use has been prospectively associated with functional impairment due to injury, illness, cognitive and emotional problems, lower health-related quality of life, and greater psychiatric symptoms [2]. Most young adults who use cannabis will do so without significant consequences, but approximately 6% of sustained cannabis users in the U.S. will develop a cannabis use disorder (CUD)—the highest rate compared to other age groups [1]. Understanding the variety of reasons for cannabis use is an important area of research and has led to examining the interaction between psychiatric symptoms and cannabis use.

## 2. Self-Medication and Young Adult Cannabis Use

Young adults frequently report using cannabis for a broad array of reasons, often arranged into five broad categories of motives for use: enhancement, social reasons, coping with mental health issues, conformity, and expansion [3]. Although these broad motives are useful to understanding and predicting patterns of use [4], the long-term effects of these use patterns [5], and foundations for interventions [6], young adults also report using cannabis specifically to cope with anxiety, depression, and other psychiatric symptoms [7,8]. It may be especially important to examine the association of cannabis use with psychiatric symptoms because young adults who use cannabis to reduce psychiatric symptoms are at increased risk of developing CUD relative to those who do not endorse this reason [9]. Drawing from prior research that demonstrated associations between heavy substance use and severe mental health problems, Khantzian (1987) theorized that for some individuals, substance use disorders emanate from a behavior known as self-medication. The self-medication theory posits that interactions between psychopathology and the specific psychotropic effects of substances of abuse contribute to some individuals’ maladaptive patterns of substance use [10]. The self-medication hypothesis holds that these individuals turn to substances of abuse to medicate themselves to reduce psychiatric symptoms and somatic symptoms [10,11]. The original theory was based on individuals who reported abuse of heroin or cocaine; however, it has since been expanded to include alcohol and other addictive drugs, focusing on the individual’s preferred substance [12]. The underlying mechanisms of self-medication can be understood within operant conditioning’s behavioral framework, specifically negative reinforcement [13,14]. For example, a person uses cannabis to reduce unpleasant psychological or physical feelings. If cannabis removes the unpleasant state or feelings, the self-medicating behavior is reinforced and will continue despite other adverse consequences, including escalating use [15]. The strength of this relationship is implicit in the substance’s reliable ability to alleviate unpleasant feelings, thus reinforcing this behavioral pattern.

Cannabis use has been examined in the context of self-medication theory [16,17,18,19], and studies have shown this behavior to be most prominent in heavier users [18,20]. Self-medication is higher in states with medical cannabis laws; however, the same states had higher self-medication before passing medical cannabis laws [21], indicating that no clear causal directionality can be established. States that have legalized “recreational” cannabis have shown increasing rates of use (15% increase compared to the national average of 11.6%), which may also include higher rates of self-medication [22]. Medical cannabis laws have been shown to increase cannabis use and cannabis use disorders among adults, and with more states passing both medical and recreational cannabis laws, young adults’ perceptions of the harmfulness of regular cannabis use have dropped [23]. This research is in its infancy, and reviews indicate a need for further exploration regarding differences in cannabis use between states where it is legal versus where it is illegal [23]. Little is known about cannabis legality and how it impacts self-medication, partly due to how new many recreational cannabis laws are in the U.S. [23].

Some gender differences have been noted in self-medication with cannabis use and young adults. Among college students aged 18 to 24, who reported weekly cannabis use, females were more likely to self-medicate with cannabis than males [24]. There is also evidence that males and females self-medicate with cannabis for differing reasons. Males more often report using cannabis to cope with social anxiety compared to females [25]. Females report more self-medication with cannabis for depressive symptoms and coping with pain [26]. Overall, self-medication is associated with higher cannabis use frequency [25], more cannabis-related problems, and greater psychological distress among young adults [27]. The current study specifically included hazardous cannabis users with CUDIT-R scores of 8 or higher [28] to provide insight into why those with hazardous cannabis use might use it for self-medication.

## 3. Psychiatric Symptoms and Self-Medication

Young adults who report self-medicating with cannabis to reduce psychological distress have been found to have higher rates of generalized anxiety [29], panic disorder [30], and obsessive-compulsive disorder [31] compared to those who report less self-medication. This behavioral pattern is also evident among young adults who self-medicate with cannabis to reduce depression symptoms. Cannabis use in young adulthood is predictive of elevated depressive symptoms, and conversely, depressive symptoms contribute to more frequent use of cannabis [32]. Findings from studies investigating causal pathways between depression and cannabis are mixed, with some studies indicating a causal link from depression to cannabis use [33] and other studies indicating that cannabis increases the risk for depression [34].

There is growing biological evidence through neurological studies of the brain’s reward and motivational systems to explain why individuals with psychiatric disorders such as depression self-medicate with cannabis [35]. In preclinical and clinical studies, there is evidence that changes in the neurotransmitters endocannabinoids (eCBs) produce antidepressant-like effects [36,37] and that the eCBs system has been implicated in the pathogenesis of depression [38]. These studies have established an association between eCBs and depression, and both appear to be linked to the reward and motivational systems. In sum, the complexity of cannabis use and psychological distress interactions is evident, prompting the need for continued research to understand the prevalence, motivations, and patterns of cannabis self-medication for psychological and physical problems in young adults.

## 4. Somatic Symptoms, Withdrawal, and Self-Medication

The use of cannabis for self-medication may also be associated with somatic and other withdrawal symptoms. Cannabis users report self-medicating to help alleviate sleep disturbances [39,40] and concentration [18]. Additionally, pain is one of the most frequently reported somatic reasons for self-medication [39,41,42]. When heavy users of cannabis want to cut back or abstain from use, they may have difficulty doing so due to mood and behavioral symptoms that vary in intensity and reflect cannabis withdrawal syndrome (CWS) [43]. Added to the DSM-V as criteria for CUD, CWS includes irritability, nervousness or anxiety, sleep difficulty, decreased appetite or weight loss, restlessness, depressed mood, and other somatic symptoms that cause discomfort [44]. These withdrawal symptoms, which overlap with the psychiatric symptoms linked to self-medication noted above, can cause functional impairment in daily activities and lead to a relapse of cannabis use [45]. CWS may be common among cannabis users, with as many as 47% experiencing this syndrome [46] and many attempting to relieve these symptoms by reinitiating or continuing cannabis use or using other substances [47,48]. Due to these linkages, the prevalence of CWS, and the high likelihood of continued use to alleviate withdrawal symptoms, it may be important to examine the associations between specific cannabis withdrawal symptoms and self-medication to inform interventions about managing the critical process of withdrawal for regular and heavy cannabis users. Examining self-medication reasons, including reasons related to CWS, in the context of hazardous cannabis use will provide new information on self-medication to relieve psychiatric and somatic symptoms as a coping strategy for hazardous cannabis users.

## 5. The Present Study

The present study was a sub-study designed to inform a large clinical trial about the prevalence of self-medication and hazardous cannabis use among young adults. The overall purpose of the study was to gather information on how young adults perceive, access, and use cannabis. The information from this study was used to inform the procedures and the selection of measures for the survey portion of the clinical trial. The study was conducted at two large public U.S. universities in moderately sized cities; one in Tennessee, where cannabis is not legal, and the other in Colorado, where cannabis is legal. While these two states have not been compared in the literature explicitly, due to the legality of both medical and recreational cannabis in Colorado and the literature reporting greater cannabis use rates in states with medical cannabis laws assessing the differences in cannabis use coping motives will add to the growing body of literature on differences in cannabis use reasons in states where laws differ greatly. Additionally, this study aims to add to the literature regarding cannabis use rates, cannabis withdrawal symptoms, and self-medication.

While previous studies have looked at self-medication with cannabis with individuals who used cannabis at various levels (including mild cannabis use), this study specifically analyzes self-medication with cannabis in a sample of individuals with likely CUD. Often self-medication is viewed in the context of various cannabis use motives. This study provides specific self-medication motives, and it is novel in providing differing reasons for self-medication (e.g., social discomfort, loneliness, and concentration) as well as a focus on mental health issues such as depressive and anxious symptoms. As previously noted, in the U.S., 6% of all young adults who use cannabis will develop a CUD. With cannabis use rates increasing during the time of the COVID-19 pandemic [49], the findings from this study can be especially relevant to aiding in prevention and treatment strategies. Having knowledge of reasons why individuals might turn to self-medication with cannabis will inform prevention and treatment and potentially lower the rates of CUD in this age group.

This study examined the likelihood of self-medicating with cannabis based on hazardous cannabis use, presence of withdrawal symptoms, state of residence, and sex—controlling for age race. Four hypotheses were tested:
**Hypotheses 1** **(H1).***Young adults who have higher hazardous cannabis use as indicated by higher CUDIT-R scores will be more likely to self-medicate*.
**Hypotheses 2** **(H2).***Young adults who report experiencing withdrawal symptoms within 72-h of cannabis discontinuation will be more likely to self-medicate than those who endorse no withdrawal symptoms*.
**Hypotheses 3** **(H3).***Participants living in Colorado, where cannabis is legal, will be more likely to self-medicate than participants living in Tennessee*.
**Hypotheses 4** **(H4).***Females will be more likely to self-medicate with cannabis than males*.


## 6. Methods

### 6.1. Procedures

All procedures were approved by the Institutional Review Board at the first author’s university and employed a reliance agreement with the second university [UTK IRB-19-05256-FB]. A certificate of confidentiality was automatically issued from the National Institutes of Health, as this was a sub-study designed to inform a larger NIH-funded clinical trial. Participants were recruited using print flyers and digital signs posted on and near the campuses. Recruitment materials included a URL link and a QR code, each of which led to a study website that provided additional details about the study before obtaining consent. Consenting participants responded to a brief online screening survey (Qualtrics, 2020) to determine eligibility for the study [50]. To ensure participants only took the survey once, the option to “Prevent Multiple Submissions” was turned on. This feature enables cookies in the web browser and prevents participants from accessing the survey more than once [50]. All IP addresses collected were removed to create a de-identified dataset for analysis. The consent let participants know that their data would be de-identified.

Eligible participants met the following criteria: (1) English-speaking; (2) young adults (ages 18–25); (3) residents of either Tennessee or Colorado; and (4) were frequent cannabis users (three or more days per week) [51] and scored eight or above on the Cannabis Use Disorder Identification Test-Revised (CUDIT-R) [25]. The CUDIT-R cut-point was used to determine those participants with likely hazardous cannabis use so that these participants would mirror those who will be enrolled in a larger, randomized clinical trial. Those who were determined to be eligible for the study completed an online survey that took less than 30 min to complete, on average. The survey contained items assessing cannabis use frequency, types of cannabis use (e.g., edibles, flower), and opinions towards cannabis use. Specifically, the survey asked participants about their cannabis use and self-medication with cannabis for psychological and physical problems, as well as any withdrawal symptoms experienced upon ceasing cannabis use. Participants who completed the online survey were compensated with an electronic gift card worth $20. The online survey was programmed with quotas to ensure that the number of respondents from each university was approximately equivalent. Recruitment proceeded between 10 January and 3 February of 2020, for a total of 3 weeks.

A total of 613 individuals clicked on the link to start the survey. One hundred and fifty-seven chose not to continue with the survey and exited it before completion. Of those who completed the survey, 290 were found to be eligible, and 166 were ineligible. All ineligible participants consented to be in the study and were fluent in English. The following number of participants were ineligible for the following reasons: four were outside of the targeted age range of 18 to 25, 58 did not live in Tennessee or Colorado, 74 used cannabis less than three days per week, and 30 had a CUDIT-R total score of less than 8. Demographic information was not collected on ineligible participants.

### 6.2. Measures

#### 6.2.1. Dependent Variables

Questions were written for this study and designed to probe specific psychiatric and somatic reasons cited in the self-medication and CWS literature to assess self-medication. Seven of the most commonly cited reasons for self-medication were selected: depression and mood [39]; feelings of isolation and loneliness [52]; anxiety and social anxiety [53,54]; sleep [40]; concentration [18]; and pain [41]. Participants were first asked, “Do you use marijuana to help any problems such as depression, anxiety, sleep problems, social stress, pain, or other issues?” The response options were “Yes”, coded as 1, and “No”, coded as 0. The endorsement of any self-medication is used as our dependent variable in the model.

If participants selected “Yes” to the dichotomous self-medication item, they were then asked, “For which problem(s) do you use marijuana for help? (Select all that apply)”. The answers were in a list with the capability of selecting multiple responses. The responses were as follows: 1. “Feeling depressed, down, or sad”, 2. “Feeling anxious, nervous, or stressed”, 3. “Problems with falling asleep, staying asleep”, 4. “Feeling uncomfortable in social situations or relationships”, 5. “Managing physical pain”, 6. “Problems concentrating or focusing on a task”, 7. “Feeling lonely or isolated” 8. “Other”, 9. “I choose not to answer”. For those who selected “Other”, an additional question asked, “You selected “other” problem. Please describe”. Participants were then able to use text entry to explain the other reasons why they used marijuana to help manage problems. The “Other” category was not treated as a variable in the model, as few participants elaborated on why they selected “Other”.

#### 6.2.2. Independent Variables

Participants were assessed for likely CUD using the CUDIT-R [28], which was a primary variable of interest in the model. The CUDIT-R is an eight-item measure of the frequency of problematic cannabis use covering four topics: consumption, cannabis-related consequences, CUD symptoms, and psychological problems. Items on the CUDIT-R were coded from zero to four and summed to form total scores with a potential range from zero to 32. Higher scores indicate more hazardous cannabis use. The CUDIT showed acceptable internal reliability with the present sample (Cronbach’s alpha = 0.70). Scores equal to or greater than 13 indicate likely cannabis use disorder, though authors of the CUDIT and more recent studies note that lower scores such as an eight can still be considered hazardous use [28,54].

Withdrawal symptoms were measured using the Survey of Acute Effects and Withdrawal [55]. In this survey, researchers identified 13 possible withdrawal symptoms (experienced within 72 h of discontinuing cannabis use). For the current study, participants were asked, “Have you ever stopped using marijuana for three or more days?” If they endorsed this item, they were asked if they have experienced any of the following withdrawal symptoms: irritability, insomnia/interrupted sleep, anxiety, loss of appetite, vivid dreams, loss of productivity, tiredness, nausea, improved productivity, weight loss, sweating, tremor, and salivation. Selecting “yes” for withdrawal symptoms was coded as 1, and “no” was coded as 0. Participants were also given the option to select, “I feel no effects when I stop using marijuana”. These participants were included in the “No withdrawal” category for the regression analyses. Secondary analyses were run using the different withdrawal symptoms as covariates predicting self-medication.

Participants’ state of residence and sex were primary independent variables of interest. Sex was coded as 0 for males and 1 for females. Participants were given the option to “choose not to answer” sex, as it was not part of the inclusion criteria. Participants who chose not to answer sex were excluded from the regression analysis, as sex was a primary variable of interest. Participants also identified their state of residence, with Tennessee coded as 0, and Colorado coded as 1. All categories coded as zero served as the reference category.

Covariates included participants’ reported age and race/ethnicity. Age was measured as a scale variable with participants reporting their age. Race/ethnicity was initially coded as a categorical variable (white, black, or African American, Native American or Pacific Islander, Asian, and the ethnicity Hispanic or Latino) with participants given the option to “Select all that apply”. As 68.8% of the sample identified as non-Hispanic white, race was dichotomized into white (coded as 0) and other/no response (coded as 1). Respondents of mixed race were also included in the “other” category. Participants who chose not to answer were also included in the “other/no response” category, as race was a covariate and not a primary variable of interest. The non-Hispanic, white majority of the sample was used as the reference category to assess any differences that may occur due to belonging to a minority race in the U.S.

### 6.3. Analytic Plan

Descriptive statistics were used to determine the prevalence of participants endorsing self-medication for any problem and the frequency of specific problems reported for self-medication and to show the sample’s demographic makeup. We also described the means and standard deviations of weekly cannabis use and CUDIT-R scores. We then tested our four hypotheses. For all hypotheses, we used a correlation test to determine the association between CUDIT-R scores and self-medicating and for each of the identified problems for self-medicating. We then tested a logistic regression model to estimate the odds of self-medication by CUDIT-R score, sex, state, and presence of withdrawal symptoms. Secondary logistic regressions were performed to establish which specific problems participants endorsed using cannabis to help. Each problem (e.g., “Feeling depressed, down, or anxious”) was a dependent variable. A final logistic regression was conducted to assess which withdrawal symptoms predicted self-medication. There were no missing values in the model that were not missing conditionally. However, participants could choose “I choose not to answer” to any survey item. Participants who chose not to answer were excluded from the regression analyses. All analytic models were conducted using IBM SPSS (2021, V28).

## 7. Results

### 7.1. Descriptive Results

A total of 290 eligible young adults responded to the online survey. Two participants selected “I choose not to answer” to the primary dependent variable and were excluded from the analysis, leaving the total sample with 288 participants. Participants ages ranged from 18 to 25 (M = 20.54, SD = 1.88). The sample of participants reported being 44.8% female and 68.8% non-Hispanic white.

Table 1 provides rates of self-medication for the entire sample. A majority of participants reported using cannabis for anxiety, sleep disturbances, and depression, while fewer participants reported using it for social discomfort and difficulty concentrating. CUDIT scores ranged from 8 to 31, with a mean score of 18.15 (SD = 5.74). Raw CUDIT scores were used to retain maximum variability. Issues of non-normality were examined, and the histogram distribution and skew (0.165) and kurtosis (−0.759) values indicated an acceptable distribution. A total of 51.7% of participants reported experiencing withdrawal symptoms after ceasing cannabis for more than three days. The most common withdrawal symptoms experienced were insomnia/interrupted sleep, vivid dreams, and irritability. The least commonly experienced withdrawal symptoms were salivation, tremors, and loss of productivity.

### 7.2. Independent Variables Correlation with Self-Medication

Correlations were run to examine the relationships between each of our independent variables and the endorsement of self-medication. Contrary to hypotheses one and three, CUDIT-R totals were negatively associated with self-medication (r = −0.126, *p* = 0.03, 95% CI: −0.24, −0.01), as was being a participant in Colorado (r = −0.121, *p* = 0.04, 95% CI: −0.23, −0.01). Supporting hypotheses two and four, endorsing withdrawal symptoms (r = 0.243, *p* < 0.001, 95% CI: 0.13, 0.35) and being female (r = 0.214, *p* < 0.001, 95% CI: 0.10, 0.32) were positive correlated with self-medicating with cannabis.

### 7.3. Predicting Self-Medication with Cannabis

A binary logistic regression was conducted to explore the relationships further and examine whether higher rates of hazardous cannabis use, sex, state of residence, and withdrawal symptoms impacted endorsing self-medication with cannabis when controlling for age and race. The assumption of the absence of multicollinearity was examined. Variance Inflation Factors (VIFs) were calculated to detect the presence of multicollinearity between predictors. VIFs greater than five are typically considered a cause for concern [56]. All predictors in the regression model have VIFs of less than five. Results of the logistic regression can be found in Table 2. Two hundred and eighty-one participants met all conditions. Seven participants chose not to answer their gender and were listwise excluded from regression analyses, as sex was a primary predictor in each analysis.

The overall model was significant (*χ^2^*(6) = 56.199, *p* < 0.001). The Hosmer and Lemeshow Chi-square was calculated to examine model fit, and the result was insignificant (*χ^2^* = 3.848, *p* = 0.871). This indicates a good model fit and does not provide evidence that the predictive values deviate from the observed probabilities in a way that the binomial distribution does not predict [57]. Supporting hypothesis two, those who reported withdrawal symptoms were 4.3 times more likely to self-medicate with cannabis compared to those who did not experience withdrawal. Supporting hypothesis four, females were 3.3 times more likely to self-medicate with cannabis compared to males. Contrary to hypotheses one and three, those with higher CUDIT-R scores were 1.1 items less likely to self-medicate with cannabis, and participants in Colorado were 2.3 times less likely to self-medicate with cannabis compared to those in Tennessee.

**Table 2 ijerph-19-01850-t002:** Results of logistic regression predicting self-medication with CUDIT-R, sex, state, and withdrawal endorsement, controlling for age and race (N = 281).

	*B*	*S.E.*	*OR*	*p*	95% C.I.
(Intercept)	−1.844	1.913	0.158	0.335	
CUDIT-R Total	−0.089	0.030	0.915	0.003	0.863, 0.970
Withdrawal Symptoms (yes = 1)	1.463	0.344	4.319	<0.001	2.202, 8.472
State (Colorado = 1)	−0.831	0.326	0.436	0.011	0.230, 0.824
Sex (female = 1)	1.190	0.341	3.289	<0.001	1.684, 6.421
Age	0.205	0.093	1.228	0.027	1.024, 1.472
Race (non-white = 1)	−0.551	0.326	0.577	0.091	0.305, 1.092

Model *x*^2^ = 56.199, *df* = 6, *p* < 0.001, Nagelkerke R Square = 0.272, −2 Log likelihood = 252.490.

### 7.4. Predicting Self-Medication with Withdrawal Symptoms

As withdrawal symptoms were a significant predictor for self-medication with cannabis, logistic regression analyses were run on the five withdrawal symptoms selected by at least 15% of the total sample: irritability, insomnia/interrupted sleep, anxiety, loss of appetite, and vivid dreams. The overall model was significant ((χ^2^(10) = 74.402, *p* < 0.001). The Hosmer and Lemeshow was insignificant (χ^2^ = 8.69, *p* = 0.369). The results of the logistic regression can be found in Table 3. There were only two withdrawal symptoms that significantly predicted self-medication. Individuals with insomnia were 7.3 times more likely to self-medicate (*p* = 0.004), and individuals with loss of appetite were 8.6 times more likely to self-medicate (*p* = 0.045).

### 7.5. Predicting Reasons for Self-Medication

All seven self-medication items were used as dependent variables to explore the significant relationships for self-medication further. This regression model only included individuals who initially selected “Yes” to the dichotomous self-medication question and again excluded listwise participants who chose not to answer their gender. Each of the original predictors was input into the new models, as were the control variables age and race. Only three models were statistically significant and fit the data well, and these were the models predicting depression, anxiety, and pain. While the model predicting social discomfort was statistically non-significant, though state of residence was a significant predictor for social discomfort. Results of these regressions can be found in Table 4. There were no significant predictors of self-medicating for sleep, isolation, or problems concentrating. CUDIT-R scores did not significantly predict specific reasons for self-medication. In line with hypothesis two, participants who endorsed experiencing withdrawal were 2.9 times more likely to self-medicate for pain (*p* = 0.001). Contrary to hypothesis three, Colorado participants were 2.4 times less likely to self-medicate for depression (*p* = 0.003), 2.5 times less likely to self-medicate for anxiety (*p* = 0.019), and 2.0 times less likely to self-medicate for social discomfort (*p* = 0.032). Additionally supporting hypothesis four, females were 3.9 times more likely to self-medicate for anxiety than males (*p* = 0.001).

## 8. Discussion

This study aimed to predict differences in self-medication with cannabis among young adults in two states with differing laws regarding cannabis legality. This study’s results add to the understanding of the varied use of cannabis among young adults to relieve psychological and physical distress. This sample can be characterized as hazardous users of cannabis who likely to meet the criteria for CUD. Given this characterization, the findings that three-quarters of the sample endorsed self-medicating, and of these, 90% reported using cannabis to help with multiple problems, suggest that these young adults are using cannabis at least in part to address their psychological, physical, and social problems. To our knowledge, this study is unique in providing a detailed description of the various types of problems for which young adults are self-medicating. While psychological problems such as anxiety and depression are often captured in general studies of motives for cannabis use, our findings reveal that a substantial percentage of young adult cannabis users are self-medicating to help with their physical problems such as sleep disturbance and pain. Another novel problem this study revealed was the endorsements of self-medicating for loneliness and social discomfort. Taken together, these descriptive findings suggest that the overwhelming majority of these young adults are self-medicating to address multiple problems.

The first hypothesis predicted CUDIT-R scores would be positively associated with self-medication with cannabis use, but results were to the contrary, showing that lower CUDIT-R scores were associated with self-medication. This finding may be interpreted that for some, self-medication is associated with more intentional and precise use of cannabis, resulting in fewer CUD symptoms. The individual CUDIT-R items were correlated with overall self-medication to explore this negative association further. The items “Not being able to stop using cannabis when started” and “Failing to do what is expected of you do to using cannabis” were both significantly negatively correlated with self-medicating. This could indicate that the young adults who use cannabis for self-medication are using it for titration and to help complete tasks, but not in a way that makes them feel as if they cannot stop or control their use. For example, if someone is using cannabis to manage anxiety symptoms, they may have greater success in accomplishing their daily tasks compared to those who use cannabis for fun. These young adults may have more self-medicating experience (e.g., understanding dose-response) and are more successful in reducing distress. With experience, they may become more efficient in their self-medicating behavior, thereby reducing hazardous cannabis use.

Secondly, as cannabis withdrawal has been added to the DSM-V as an indicator of CUD, it was important to assess whether withdrawal symptoms lead to self-medication. Those who experienced withdrawal symptoms were more likely to self-medicate with cannabis, than those who experienced no withdrawal symptoms. Nearly a quarter of the sample endorsed experiencing at least one withdrawal symptom. As withdrawal symptoms can overlap with and intensify self-medication reasons (e.g., anxiety, sleep disturbance, and depression), we ran further analysis. While the dichotomous “Do you experience withdrawal symptoms?” significantly predicted self-medication for pain, only the withdrawal symptoms of insomnia and loss of appetite predicted the dichotomous endorsement of self-medication. This supports that individuals in this sample are able to differentiate between symptoms related to their cannabis cessation and symptoms that they experience that are potentially unrelated to cannabis. These data support withdrawal as a crucial time for intervention, as these symptoms can lead to functional impairment for the individual, leading to increased and continued cannabis use [45].

Our third hypothesis was also not supported and was contrary to what was seen in the literature. Participants in Colorado self-medicated with cannabis less than those in Tennessee. Tennessee participants reported self-medicating for depression, anxiety, and social discomfort. While no directionality had previously been established on the legality of cannabis and self-medication [22], these results pose interesting findings for the current study. Research has posed that harm reduction can be both a social and biological construct [58]. This could mean that participants in Colorado might have greater availability of cannabis due to the legality. They may not see using cannabis as helping with feelings of anxiety, isolation, and social discomfort as it is more a part of everyday life. Another possible explanation could be that for those in states where medical cannabis is legal, participants are getting prescribed cannabis for psychological and somatic concerns, lowering the amount of self-medication. Research has found that in states with legalized medical cannabis, there has been a drop in prescriptions for psychiatric and somatic conditions [59]. Recent research has provided evidence supporting using cannabis and cannabinoids for psychiatric conditions; however, this research is in its infancy, and systematic reviews indicate a need for more exploration [60]. With Colorado having medical cannabis laws in place, it may be that participants are not feeling the need to self-medicate because they can receive cannabis as a form of medication. Future research on state differences in self-medication should seek to determine whether those who are prescribed medical cannabis consume more than prescribed to achieve a reduction in psychiatric symptoms or whether they use cannabis in place of or to enhance the effects of other psychiatric medications.

Finally, the results that females are over three times as likely to endorse self-medication supports our final hypothesis and other related research [24]. Interestingly, sex did not predicate self-medication for depression or pain, which was previously found in the literature [26]. However, the results that females are over three times as likely to endorse self-medication supports our final hypothesis and other related research [24]. It is not surprising that females endorsed self-medication for anxiety at greater rates (threefold) than males, given that females are 64% more likely to have any anxiety disorder relative to males, with rates at 23.4% versus 14.3%, respectively [61,62]. It is not clear from the current study whether cannabis is being used exclusively, as a substitute, or in combination with formal anxiety treatment. Because psychiatric disorders were not assessed in this study, it is also unclear if the participants’ interpretation of their anxiety or depressive symptoms would meet the American Psychiatric Association’s Diagnostic and Statistical Manual (5th Edition)’s criteria for generalized anxiety disorder, other related anxiety diagnoses, or depressive episodes.

Results need to be interpreted in light of the study limitations. First, this study was conducted prior to the beginning of the impacts of COVID-19 in the U.S. In the U.S., both cannabis use and mental health symptomology have increased in the context of the global pandemic, where lockdowns and isolation have been prevalent to promote public health and prevent the spread of disease [49]. The results might differ if this same study were conducted following the pandemic and related lockdowns. Additionally, as a cross-sectional study, interpretations are limited to a single time point. Follow-up research with this sample could provide more detail on the interaction between cannabis use, psychological and physical distress, and time, capturing developmental vicissitudes. The sample size was limited and only considered young adults with frequent cannabis use. With participants given the option the “choose not to answer” for the sex item, leading seven participants to be excluded from the analyses, the sample size for the regression models dropped to 281 and 214, respectively. Further assessments should include less frequent (i.e., 1 or 2 days per week) cannabis users to examine whether they have any endorsement of self-medication. It is possible for young adults without regular or hazardous cannabis use to self-medicate. Young adults with any cannabis use have often been the population examined with the self-medication theory and cannabis use [21], and therefore these results can only be generalized to those with hazardous cannabis use. Further, because the original study design was a descriptive investigation to inform a clinical trial, the assessment battery did not include detailed mental health measures. In particular, knowing more about participants’ mental health could be helpful in conjunction with the self-medication framework. Knowing about participants’ mental health could have allowed for a more robust measure of self-medication, eliminating the dichotomized variable. Finally, the sample was primarily white college students, limiting the generalization of findings beyond this group’s characteristics and behavioral profiles. These participants were also located in only two states, limiting the generalizability to states across the US. This also does not allow for international comparison. The study also required adults to be English speaking and therefore cannot account for individuals who do not speak English. Additionally, participants, especially those in Tennessee, were willing to take a survey on cannabis use, indicating that they did not have serious privacy concerns related to their cannabis use and its legality.

## 9. Conclusions

The primary contributions to the literature from the current study are threefold: (1) the assessment of self-medication among young adults who likely meet CUD criteria, (2) the detailed examination of the types of psychological, physical, and social problems; and (3) the analysis of these results by sex, state, and withdrawal symptoms. Providing comprehensive assessments that include psychological, social, and physical distress in conjunction with understanding the unique features of an individual’s cannabis use behaviors could offer clinically meaningful intervention targets for young adults. The knowledge that withdrawal increases the likelihood of self-medication can initiative a new time-point for interventions for individuals wanting to cease or reduce their cannabis use. Additionally, the knowledge of reasons for self-medication with cannabis in states where cannabis is not legal can guide legislators on future and current medical cannabis laws. Based upon these initial findings, specific research on the utility of simultaneously addressing CUD, self-medication, and psychological and physical distress is warranted.

## Figures and Tables

**Table 1 ijerph-19-01850-t001:** Descriptive statistics.

	Total (*n* = 288)
Self-medication	
No	24.0% (69)
Yes	76.0% (219)
For which problems do you use marijuana? *	
Depression	59.4% (130)
Anxiety	81.7% (179)
Sleep Problems	79.0% (173)
Pain	40.6% (89)
Loneliness or Isolation	37.9% (83)
Social Discomfort	32.0% (70)
Concentration	19.2% (42)
CUDIT-R Total Score	M = 18.15, SD = 5.74
Withdrawal Symptoms	
No	42.8% (124)
Yes	51.7% (150)
No cannabis cessation for three or more days	5.5% (16)
Which withdrawal symptoms do you experience? *	
Irritability	18.4% (54)
Insomnia/Interrupted Sleep	23.3% (67)
Anxiety	15.4% (44)
Loss of Appetite	14.6% (42)
Vivid Dreams	20.1% (58)
Loss of Productivity	2.1% (6)
Tiredness	5.2% (15)
Nausea	5.6% (16)
Improved Productivity	7.6% (22)
Weight Loss	3.5% (10)
Sweating	3.8% (11)
Tremor	1.4% (4)
Salivation	0.7% (2)
State of Residence	
Tennessee	51.4% (148)
Colorado	48.6% (140)
Sex	
Male	52.8% (154)
Female	44.8% (129)
I choose not to answer	2.4% (7)
Age	M = 20.54, SD = 1.88
Race	
White	68.8% (193)
Other/Not Answered	31.3% (90)

* Measured with a single “select all that apply” item.

**Table 3 ijerph-19-01850-t003:** Results of logistic regression predicting self-medication with CUDIT-R, sex, state, and specific withdrawal symptoms, controlling for age and race. (N = 281).

	*B*	*S.E.*	*OR*	*p*	95% C.I.
(Intercept)	−1.706	1.965	0.182	0.385	
CUDIT-R Total	−0.091	0.031	0.913	0.003	0.860, 0.969
Withdrawal Symptoms					
Irritability (yes = 1)	−0.292	0.525	0.747	0.579	0.267, 2.091
Insomnia (yes = 1)	1.984	0.690	7.273	0.004	1.882, 28.112
Anxiety (yes = 1)	1.998	1.171	7.371	0.088	0.743, 73.009
Loss of appetite (yes = 1)	2.157	1.074	8.649	0.045	1.054, 70.998
Vivid dreams (yes = 1)	0.355	0.474	1.426	0.454	0.631, 3.614
State (Colorado = 1)	−0.803	0.337	0.448	0.017	0.231, 0.867
Sex (female = 1)	1.006	0.352	2.735	0.004	1.370, 5.456
Age	0.208	0.095	1.232	0.028	1.022, 1.484
Race (non-white = 1)	−0.607	0.336	0.545	0.070	0.282, 1.052

Model *x^2^* = 74.402, *df* = 10, *p* < 0.001, Nagelkerke R Square = 0.349, −2 Log likelihood = 234.286.

**Table 4 ijerph-19-01850-t004:** Results of logistic regression predicting self-medication reasons with CUDIT-R, sex, state, withdrawal endorsement, controlling for age and race. (N = 214).

Dependent Variable	*B*	*S.E.*	*OR*	*p*	95% CI
Depression					
(Intercept)	−0.672	1.673	0.688	0.688	
CUDIT-R Total	0.056	0.029	1.058	0.051	1.000, 1.120
Withdrawal Symptoms (yes = 1)	−0.273	0.316	0.761	0.388	0.409, 1.414
State (Colorado = 1)	−0.872	0.296	0.418	0.003	0.234, 0.747
Sex (female = 1)	0.368	0.293	1.444	0.209	0.814, 2.564
Age	0.024	0.077	1.024	0.759	0.881, 1.190
Race (non-white = 1)	−0.320	0.328	0.726	0.330	0.382, 1.382
Model χ^2^ = 15.847, *df* = 6, *p* = 0.015, Nagelkerke R Square = 0.096, −2 Log Likelihood = 274.735					
Anxiety					
(Intercept)	−1.687	2.244	0.185	0.452	
CUDIT-R Total	0.008	0.037	1.008	0.824	0.937, 1.084
Withdrawal Symptoms (yes = 1)	0.254	0.406	1.289	0.533	0.581, 2.858
State (Colorado = 1)	−0.953	0.389	0.386	0.014	0.180, 0.827
Sex (female = 1)	1.305	0.399	3.687	0.001	1.686, 8.065
Age	0.139	0.105	1.149	0.183	0.936, 1.410
Race (non-white = 1)	−0.109	0.414	0.897	0.792	0.398, 2.020
Model χ^2^ = 18.949, *df* = 6, *p* = 0.004, Nagelkerke R Square = 0.138, −2 Log Likelihood = 184.256					
Pain					
(Intercept)	−4.848	1.723	0.008	0.005	
CUDIT-R Total	−0.051	0.029	0.950	0.081	0.898, 1.006
Withdrawal Symptoms (yes = 1)	1.077	0.337	2.936	0.001	1.516, 5.685
State (Colorado = 1)	−0.345	0.306	0.708	0.259	0.389, 1.290
Sex (female = 1)	−0.264	0.299	0.768	0.376	0.428, 1.379
Age	0.242	0.080	1.274	0.002	1.090, 1.490
Race (non-white = 1)	−0.202	0.346	0.817	0.556	0.416, 1.603
Model χ^2^ = 22.143, *df* = 6, *p* = 0.001, Nagelkerke R Square = 0.133, −2 Log Likelihood = 266.227					
Social Discomfort					
(Intercept)	−4.262	1.757	0.014	0.015	
CUDIT-R Total	0.050	0.029	1.051	0.088	0.993, 1.113
Withdrawal Symptoms (yes = 1)	−0.045	0.330	0.956	0.892	0.500, 1.826
State (Colorado = 1)	−0.679	0.317	0.507	0.032	0.272, 0.944
Sex (female = 1)	0.286	0.306	1.331	0.349	0.731, 2.424
Age	0.132	0.080	1.141	0.100	0.975, 1.334
Race (non-white = 1)	0.133	0.346	1.143	0.700	0.580, 2.251

Model χ^2^ = 10.568, *df* = 6, *p* = 0.102, Nagelkerke R Square = 0.068, −2 Log Likelihood = 256.993.

## Data Availability

Data are part of a parent study which is ongoing and therefore cannot be made available at this time.

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
