# Peer review of "Predicting Self-Medication with Cannabis in Young Adults with Hazardous Cannabis Use"

_ijerph, 2022, doi:10.3390/ijerph19031850_

Round 1

Reviewer 1 Report

Thank you for providing me with the opportunity to read this revised paper. Seems that some work has been done in order to improve it. However, this is still insufficient, and many parts of the article keep suffering from many pitfalls which prevent this manuscript from being published in the current form. These pitfalls again refer to the literature, methodology and presentation and interpretation of the findings. Many things in it still sound unclear, incomplete, misleading or just odd.

However, since everything is quite doable, I would like the authors to undergo another major revision.

I outline my comments for each part in the attached file. Again, everywhere I ask you to provide/explain/clarify/elaborate or something similar, please do it in the text first and then report on how you related to each comment.

Author Response

Dear Reviewer,

We thank you for your time and feedback on our manuscript. We have revised the manuscript to address each of the reviewers’ concerns, and we detail those revisions below. Below we provide the point-by-point responses. All modifications in the manuscript have been documented with track-changes. We believe the revised manuscript more clearly reports our findings, and we look forward to hearing back from you.

Reviewer 2 Report

Now the manuscript is clear and the conclusions are  consistent with the evidence and arguments presented

Author Response

Dear Reviewer,

We appreciate your time and feedback on our manuscript. Thank you. 

Round 2

Reviewer 1 Report

Thank you for the very good job done with respect to responding to my previous comments. The article looks much better and much more ordered and sounding. However, there are still issues that must be treated with care and attention. Given that the review has already undergone two rounds, I would like this to be the last one. Please take the revision of the article even more seriously than in the previous rounds.

In any case, what I would like to warmly suggest is a self-proofread. You can do it before relating to my comments or after it – it does not matter. This will prevent typos, illogical or awkward expressions, unnecessary long sentences, logical errors, and improve the language and the appearance of the article. I do it every time before I submit my new manuscripts even for proofreading (not to talk about the submissions to journals).

Abstract

- Please amend the typo in "were cannabis is legal". It is "where cannabis is legal".

- The models examining self-medication for social discomfort and sleep problems as reasons for self-medication are non-significant. Therefore, please remove the findings related to them. I will refer to the non-significant models later in the Results.

- In the keywords, you can consider adding the words like "anxiety" and "withdrawal symptoms". They are much more salient in light of your findings than, for example, "cannabis use".  

Introduction

- In my previous comment, I asked you to put the heading. Maybe I was unclear, but I meant to put the word "Introduction", not the title of the article again. Please perform that.

Self-medication and young adult cannabis use

- Lines 64-65 - You can just write "coping with mental health issues".

- Lines 104-107 – what you have provided there still leads me to argue that the sex-related hypothesis should be formulated as two-tailed, or there should be a sub-hypothesis which will address the particularities in the mentioned gender differences.

The present study

- Line 175 - you do not control for state of residence as this is one of your predictors. Please move it nearby sex.  

- As to H1, I did not understand what do you mean by the "higher rates of hazardous cannabis use". Why not to say "greater use disorder" or something similar and more simple? Please rewrite.

- In H2 you have left the sentence "Females will be more likely to self-medicate with cannabis than males". Please remove.

- Lines 182-183. I do not quite understand the meaning of the sentence. In any case, the hypotheses are tested as two-way unless you examine correlations. Please reconsider this sentence.

- I have previously mentioned that this subsection does not contain any information about the disadvantages of the previous research on self-medication. I saw your response, but I am still not convinced by it. Again, I want you to provide this information while keeping the following questions in mind: In which aspects does this article do a better job that the previous ones? Which information does it provide that the others did not manage to provide? If this requires further literature review then please do it.

- I have previously asked you to mention the theoretical and the empirical contributions of this study to the field. Your response has partially provided an answer to this question. Please elaborate more here.

- Line 171 – which "gaps" do you mean? Please be more specific.

- I have previously asked you to mention how this study can be relevant for the pandemic? I saw your response, but unfortunately cannot agree with what you have maintained. It does not matter when the research has been conducted. It is about the applicability and relevance of the findings for any period, any situation, any setting, any context. The findings of each study do not exist in a vacuum and are not suited for any particular case or any particular period but rather reflect some broader phenomenon which should be discussed in the broad academic setting regardless of the study conduction period. This is especially important with respect to the research which can be highly relevant for the period when it is expected to be published. As known, substance use has undergone changes in many countries during the pandemic because of elevated mental responses to it. Hence, what you have found may be salient also nowadays and, to some extent, even provide solutions for the current situation. Please justify the relevance of the findings for the current period.

Methods

Procedures

- What do you mean by "outside of age range". Meaning what ages? Please specify.

- I have previously asked you to provide the source stating that frequent use is three times a week at least. I did not find this information in the source which you have provided. Can you please specify?

Measures

- Line 249 – the disorder symptoms variable was not a covariate. Your only covariates are age and race. Please remove this part of the sentence.

- Line 271 – in line with the previous comment – you have a clear distinction between the role of sex and state and that of age and race. Therefore, you cannot put them under one heading. Please separate.

Results

- Line 305 – you can just write "67%".

- It is still unclear how did you treat the "I choose not to answer" category in the sex variable. Please elaborate. If they were defined as missing cases, then your analytical sample basically consists of 283 (or even 281) and not 288 cases.

- Concerning the race variable – it is unclear why the very well-defined group of non-Hispanic Whites has been defined as the reference category.

- You did not refer to the descriptive statistics for withdrawal symptoms. Please provide them when describing the data shown in Table 1.

- Please change the emphasis style in Table 1. Please italicize the names of the categories, not variables.

- Line 323 – the heading for the paragraph is quite long. You have already mentioned that you use logistic regression, so there is no need in tautology. Please provide something like "Predicting the self-medication". Same comment is with respect to the headings in lines 347, 359, and 379.

- Line 335 – you can present the results of the Hosmer and Lemeshow test exactly like in the case of model significance. Same is for what you wrote in line 351.

- Lines 338-340 – the sentence is very long. Please split it, and mention the reference group in each case (more likely than whom?).

- Line 345 – the words "predicting self-medication should come after the word "regression". In addition, there is no need to start each word with the capital letter in table headings.

- Table 2 – since except for CUDIT score and age, all your variables are dummies, please write in the brackets for each one what constitutes the "1" category (for instance: Sex [female = 1]). Same comment is regarding the other tables.

- Lines 354 – please write "7.3 times" and not "7.3x". Same is about all the rest of similar cases throughout the section.

- Lines 365-366. Unfortunately for you, this is a borderline significant result which should in any case be referred to as non-significant. Please rewrite or remove this part of the sentence.

- Throughout the section, you use very vague expressions like "our hypothesis" or "our initial hypotheses". The purpose of the early numeration of the hypotheses is to avoid this unclear way of expressing yourself. Please rewrite using the expressions like "Contrary to Hypothesis 3…" "In line with Hypothesis 4…".

- Now – concerning what I have promised in the Abstract. You have provided the results of all models in Table 4 and 5 without considering the basic fact that most of them are non-significant. Such models have absolutely no meaning (as entering the predictors in Block 1 does not significantly reduces the error as compared to Block 0 where the dependent variables predict themselves by themselves), so there is also no need to refer to the significant findings inside them. Therefore, please eliminate these models from the tables and remain with the significant ones only. Please also ensure that each one of them fits the data well. Given your small sample size, this should not be taken for granted. If some of the significant models do not fit well to the data, again, you cannot use them and report their findings. You should also mention that you ran seven models but only some of them were significant and fitted the data well. Considering this, please remove the parts in which you referred to the findings on social discomfort and sleep problems.

- Considering Table 5 – after looking at it for another time – I have doubts concerning the need for this analysis. It does not provide any significant information. And again, since the models predicting social discomfort and sleep problems as the reasons for self-medication are non-significant, there is no meaning for the significant coefficients in them. The rest of the findings resemble those from the previous analysis.

- Throughout the section, and in previous sections, you overuse the word "endorse" (in any tense). Please replace it in at least part of the cases since it falls out of context in many sentences.

Discussion

This part should undergo a major revision as the language used there and the way of interpreting the findings sound very distant from the academic style and sometimes borders with amateurism:

  1. The section should start with some reminding on what the goal of the study was, what did it intend to find or make new.
  2. The tautology should be avoided, like in Line 412.
  3. The hypothesis, before predicting something, assumes some relationship. Therefore, the word "predicted" in Line 413 should be replaced.
  4. Seems that you do not follow the direction of the associations between the variables. This is clearly seen in Lines 415-417. Please amend.
  5. You should incorporate the word "whether" much more in your academic life. Believe me, this will assist you a lot there. Please replace the word "if" with this one in Lines 430, 464, and 475.
  6. Please do not leave something unexplained. For example, the criteria for what should the participants meet in line 478? Please be more specific. Same is about what this study adds to understanding of what you wrote in Lines 397-398.
  7. Please use a much more mature language. Avoid expressions like "it is not unexpected" (double negation), "achieve a homeostasis" (you do not know whether they indeed do), "as hazardous or problematic users" (they should be either these or those),
  8. You do not have to duplicate the Results section here. Therefore, please remove the statistical findings (like "three times", "being 1.31").
  9. Please remove all the discussion of findings solicited from the non-significant models (Lines 434-444 and elsewhere).
  10. Please go through the section and amend the typos.

Author Response

This manuscript is a resubmission of an earlier submission. The following is a list of the peer review reports and author responses from that submission.

Round 1

Reviewer 1 Report

Thank you for providing me with the opportunity to review this article. It is quite interesting and may provide some newer information on the current tendencies in cannabis self-medication, especially due to the fact that you try to explain it using two dimensions which are domains of research by themselves – cannabis use disorder and cannabis withdrawal. However, the road is quite long until this article will be considered as suitable for publication, as it suffers from some pitfalls that must be thoroughly addressed.

The major concern with the article was the chosen analytical strategy. It is unnecessarily oversophisticated is some places and undersophisticated in the others and does not address things simultaneously. Some of the analyses are quite unnecessary as they do not refer to your dependent variable which is self-medication. The analyses are performed while controlling for a limited number of variables, which is quite meaningless in light of the fact that so many aspects are addressed in one study. Therefore, I am very much unsure that once all relevant variables of interest are entered into one statistical model, the findings would remain the same. Moreover, the change of the variable role throughout the analysis (as in case of gender) is inappropriate. Same variables cannot be variables of interest and control in one study. Furthermore, many of the variables seem not to be thoroughly examined (as in case of CUDIT-R score). It is very much likely that performance of analyses, where these variables are presented in another way, would yield quite different results. The authors should redo the analysis completely in accordance with my comments below.

I also provide numerous comments for each part of the article except for Discussion, as it must be revised following the revision of the analyses.

Title

Please change it because it does not cover everything you examined in the study.

Abstract

- It is unnecessarily long as things can be presented in a much more straightforward way.

- You wrote "and over half reported daily cannabis use (56.2%)". You can just write "and 56.2% reported daily cannabis use". Please provide the figure in the following sentence in a similar vein.

- You wrote "(scores > 13 = likely cannabis use disorder)". This information is unnecessary at this stage. Please remove it.

- Please remove odds ratios and other results of analysis. Abstract is not the place to report them.

- Please remove the following sentence "For this sample, as CUDIT-R scores increase by one unit, the odds of endorsing self-medication decreased by a factor of .947." The findings section is the only proper place for sentences like this one.

Self-Medication and Young Adult Cannabis Use

- You wrote "The self-medication hypothesis holds that these individuals turn to substances of abuse to medicate themselves to reduce psychiatric symptoms." First, what are these substances (it can be assumed that any substance may be defined as such). Second, is the self-medication performed only for the mentioned purpose? What about treatment of health conditions, such as chronic pain? Please elaborate more here.

The present study

- You should stay with hypotheses only. There is absolutely no need in their duplication by research questions, especially in light of the fact that all your hypotheses are one-tailed. Basically, you have one research question which refers to examination of factors associated with self-medication. This is how you should formulate it in the text. Please do all this.

- Please revise the formulations of your hypotheses in accordance with the analyses that you perform in order to test them.

- I would also suggest to put the hypotheses following the literature review which will better justify their formulation in a way you choose. For that purpose, you should provide a deeper literature review, especially on association of gender and state of residence. Please elaborate.

- Please numerate the hypotheses (H1, H2, and so on). This will allow you to significantly reduce the wording in both the Analytic Plan subsection and in the Results section as there will be no need to repeat their full formulation.

- Please ensure that all your hypotheses can be examined using inferential analyses. There is no need for hypothesizing something that can be shown using descriptive statistics.

- Please include the variable differentiating between the participants whose data were collected in January and those in February. The reasons for self-medication may significantly differ between these months as the situation with COVID-19 became worse in February 2020. Numerous studies conducted in the early COVID-19 period (even before it was defined as pandemic by the WHO) point at elevated emotional and mental reactions to the situation (numerous Chinese studies can be used here); thing that may lead to increased substance use. Please use it as a control variable.

Methods

- Please provide the numbers of all relevant certificates obtained for the study.

- What about the IP addresses? Did you eliminate them? Please explain and also say whether the information about the non-storage of IP addresses was included in the consent for participation. Please elaborate here.

- Did you ensure that there were no duplicate observations in your database? If yes, how? Please elaborate.

- In general, please provide more information on the study. Was it a part of larger research? Was it a follow-up or, on the contrary, pilot study for some other/more general project? What was the general purpose of conducting it?

- Did the questionnaire include only the items that were used for definition of the study variables or were there additional items? Please explain.

- You wrote "Eligible participants were English-speaking young adults, ages 18 to 25, who were frequent cannabis users (three or more days per week) and had scores of 8 or above on the Cannabis Use Disorder Identification Test-Revised (CUDIT-R) [24]." Were these the only eligibility criteria, or there were some others? Please elaborate.

- How were the missing values handled? Please explain.

- Please state clearly that self-medication is your dependent variable. Please also separate between the main predictors and covariates using proper subheadings.

- How was the age measured? Please provide.

- What were the original categories of race? How is it operationalized in the study? What is the reference category? Please elaborate.

- I am more than sure that the frequency of cannabis use was skewed. Please examine the normality of its distribution. If it is indeed skewed, please dichotomize it in accordance with Sznitman (2017). Do it only if you decide to use this variable also for inferential analysis. If you do not want to include it there, please leave it as a part of CUDIT-R.

- You mentioned the bounds for certain diagnoses that reflect certain scores in CUDIT-R. First, your total score variable should be reexamined as it again may be skewed. If yes, then you must apply the categorization you mentioned (>13 and 8-13, which will serve as a reference category).

- You mentioned that there was a further examination of self-medication-related problems. It is quite clear that this variable should be analyzed separately. In any case, you should include it in your secondary analyses.

- How did you treat the "other" category? As a variable? What was the reason to include it?

- Same as above applies to withdrawal symptoms. Similar to problems, this should go to the secondary analysis as its examination refers only to people who reported having at least one withdrawal syndrome.

In sum:

  1. Your primary analysis should deal with the estimation of the likelihood of the cannabis self-medicating. You predict this with the following variables: sex, state, presence of CUD (may be performed as sensitivity analysis in case the continuous variable is not skewed), and presence of withdrawal symptoms (yes/no). This way, you already test four hypotheses. You should also enter here your control variables: age, race, and the month of data collection. You may enter all variables in one step or examine four models by adding each independent variable (while controlling for the covariates) at each step.
  2. Only then you should perform your secondary analyses on differences in problems and withdrawal symptoms. In any case, they should be examined in relation to self-medication and not the other variables. Again, this is because self-medication is the phenomena that you study.
  3. Please perform collinearity diagnostics in your logistic analysis.
  4. Please perform Hosmer & Lemeshow goodness-of-fit test and report its result. If it demonstrates poor fit, please reconsider the model.

Tables

- Please provide better heading for each table

Table 1:

- This table must contain descriptive statistics for all the studied measures and covariates. Please elaborate.

- Please remove the "N=290".

Table 2 should include the results of the logistic regression analysis in accordance with the abovementioned plan. In addition to model chi-square result, and Nagelkerke R, please report the valid N and -2log likelihood value.

The subsequent tables should include the results of your secondary analyses.

Limitations

You have much more limitations than you have outlined. First, this sample included only those young adults who did not have serious privacy concerns. I am more than sure that there are lots of young adults who self-medicate cannabis in the U.S. However, due to sensitivity to this topic, they might avoid participating in similar studies in order not to expose themselves and discover their cannabis use patterns. Second, the samples are restricted to the two U.S. states. Third, in general, the study design does not allow international comparison. Therefore, it is unknown whether same factors explain self-medication in the EU, Israel, Australia or any other countries where cannabis use is a highly studied topic. Fourth, your sample was quite small. Fifth, self-medication was presented dichotomously, thereby restricting your analytical options. Sixth, your sample included English-speaking young adults only, frequent cannabis users and those who have some cannabis use disorder or potential for disorder. This specification of the sample restricts the generalization of your findings on the entire population of young adults who consume cannabis, and even those who do it for self-medication. Also those who speak other languages, consume cannabis less frequently, and have no CUD or no potential for it can self-medicate cannabis. Please add all these to the text.

Reviewer 2 Report

The aim of the study is to examine the prevalence of self-medication and problematic cannabis use among young adults.  

The topic is of interest. The manuscript is updated (although bibliography needs to be revised). 

Some points need to be clarified or improved. 

We attach a file in which we explain the details.

Reviewer 3 Report

The current cross sectional study examines relationships between cannabis use frequency, self-medication, gender, and state in a sample of hazardous cannabis users. The study has notable strengths, including a large sample size and a comprehensive assessment of reasons for self-medication across many different problems. The description of self-medication in the introduction is comprehensive. However, the manuscript also has some limitations:

  • Given that this is a cross-sectional survey, findings are not especially novel.
  • Implications of these findings could be better developed. For instance, what are the implications that most (76%) of young adults with hazardous cannabis use are self-medicating for multiple problems? Are there implications for treatment interventions or policy?
  • In paragraph 2 of the discussion, the authors comment that Colorado residents are more likely to self-medicate with cannabis. However, this is not true when controlling for sex and age, as discussed in a later paragraph (beginning line 411).
  • An important limitation to mention is that this is a sample of hazardous cannabis users and therefore results may not be generalizable to all young adult cannabis users. This also is important to consider when comparing findings to other studies. For instance, in line 414-15 the authors state that finding Colorado participants to be less likely to self-medicate is “contrary to what is stated in the literature”, but the study cited (19) is a very different sample (general adults vs. young adult hazardous users in the current study). This is an opportunity to consider why these different populations may differ in likelihood of self-medicating with cannabis.
